



# A unified code for conventional and disjunct eddy covariance analysis of trace gas measurements: An urban test case

Marcus Striednig[1,2], Martin Graus[1], Tilmann Märk[2], Thomas G. Karl[1]

[1]ACINN (Institute of Atmospheric and Cryospheric Science) Leopold-Franzens University Innsbruck, Austria
[2]Institute for Ion- and Applied Physics, Leopold-Franzens University Innsbruck, Austria

*Correspondence to*: Thomas Karl (thomas.karl@uibk.ac.at)

**Abstract.** We describe and test a new versatile software tool for processing eddy covariance and disjunct eddy covariance data. We present an evaluation based on urban NMVOC measurements using a Proton-transfer-reaction quadrupole interface time of flight mass spectrometer (PTR-QiTOFMS) at the Innsbruck Atmospheric Observatory. The code is based on MATLAB ® and can be easily configured to process high frequency, low frequency and disjunct data. It can be applied to a wide range of analytical setups for NMVOC as well as other trace gas measurements, and is tailored towards the application of noisy data, where lag-time corrections become challenging. Several corrections and quality control routines are implemented to obtain the most reliable results. The software is open-source, so it can be extended and adjusted to specific purposes. We demonstrate the capabilities of the code based on a large urban dataset collected in Innsbruck, Austria, where ambient concentrations of non-methane volatile organic compounds (NMVOC) and auxiliary trace gases were sampled with high temporal resolution above an urban canopy. Concomitant measurements of $^{12}$C and $^{13}$C isotopic NMVOC fluxes allow testing algorithms used for determinations of flux LODs and lag time analysis. We use the high frequency NMVOC data set to generate a set of disjunct data and compare these results with the true eddy covariance method. The presented analysis allows testing the theory of DEC in an urban environment. Our findings confirm that the disjunct eddy covariance method can be a reliable tool, even in complex urban environments, when fast sensors are not available, but that the increase in random error impedes the ability to detect small fluxes due to higher flux LODs.

## 1 Introduction

Eddy covariance (EC) is the method of choice for most micrometeorological studies of surface fluxes (e.g. Dabberdt et al., 1993; Aubinet et al., 2012). It has been extensively used in atmospheric sciences (e.g. Horst et al., 2004; Oncley et al., 2007; Foken et al., 2010; Patton et al., 2011) and biogeochemistry (e.g. Ameriflux: https://ameriflux.lbl.gov/; Euroflux: http://www.europe-fluxdata.eu/icos; Baldocchi et al., 1988; Fowler et al., 2009; Aubinet et al. 2012; Rannik et al, 2012; Ducker et al., 2018). The use of EC for atmosphere-surface exchange measurements is widespread and a number of commercial, freely distributed closed and open source codes for the analysis of EC data are available (e.g. Fratini et al., 2014; Mauder et al., 2008, Metzger et al. 2017).



The basic concept of the eddy covariance method is derived from the budget equation after Reynolds decomposition (e.g. Stull, 1988):

$$\frac{d\langle c\rangle}{dt} + \frac{d\langle w'c'\rangle}{dz} + \langle u\rangle\frac{d\langle c\rangle}{dx} + \langle v\rangle\frac{d\langle c\rangle}{dy} = \langle Q_S\rangle, \qquad (1)$$

where the first term represents the storage component of a tracer of concentration $c$, the second term is the vertical flux divergence and the third plus fourth terms represent horizontal advection. Brackets denote temporal averages and primes the turbulent fluctuation of a quantity. These terms must balance all sources and sinks ($Q_S$) in the control volume. If advection terms in the vertical dimension can be considered negligible, the driving part for vertical exchange is given by the second term as the covariance between vertical wind (w') and tracer (c') fluctuations. The vertical turbulent flux in the atmosphere can be defined as:

$$F_S = \langle w'c'\rangle \qquad (2)$$

In the past EC has been largely restricted to a limited number of species (e.g. H2O, CO2, CH4) due to the requirement of fast sensors (ideally sampling frequencies > 10Hz). For reactive trace gases (such as non-methane volatile organic compounds: NMVOC), that participate in air chemistry, instruments capable of EC have been much more limited (Guenther and Hills, 1998, Karl et al., 2001). However, owing to advances in mass spectrometry (Hansel et al., 1995; de Gouw et al., 2003;

Jordan et al., 2009; Graus et al., 2010; Blomquist et al., 2010) and optical techniques (Kroon et al., 2007; Ammann et al., 2012; di Gangi et al., 2011; Muller et al., 2009) EC for reactive trace gases has recently become more tractable. A number of studies have used these new techniques to investigate emission and deposition processes of reactive gases (e.g. Karl et al., 2001, 2002, 2010; Velasco et al., 2009; Langford et al., 2009; Ruuskanen et al., 2011; Park et al., 2013; Nguyen et al., 2015) and aerosols (e.g. Nemitz et al., 2008; Farmer et al., 2013; Deventer et al., 2015). A technological milestone in atmospheric

sciences for the analysis of trace gases and aerosols has been achieved in the last couple of years through the introduction of time of flight mass spectrometers (TOF-MS) (DeCarlo et al., 2006; Jordan et al., 2009; Graus et al., 2010). Chemical ionization methods coupled to time of flight mass spectrometers are becoming sensitive enough to simultaneously measure a wide range of minute amounts of trace gases and aerosols fluxes. TOF-MS inherently obtain all mass channels of each spectrum virtually simultaneously and are therefore capable of true EC measurements of a wide range of species (Müller et

al., 2010; Kaser et al., 2013; Park et al., 2013).

Noteworthy, there is a variation of EC, called disjunct eddy covariance (DEC) (e.g. see review by Rinne et al., 2012), which was originally based on the intermittent sampling strategy explored for in-situ aircraft measurements of trace gases (Dabbert et al., 1993; Lenschow et al., 1995; Cooper and Shertz, 1995). Here an air-sample is captured physically fast enough (e.g. at 0.1s) with a disjunct eddy sampler (DES) so that it still contains the information about turbulent fluctuations within the air

mass. A first realization of a DES sampler (Cooper and Shertz, 1995) was tested on an aircraft. Subsequent improvements, allowing to intermittently store and analyze trace gases faster, were first implemented by Rinne et al., 2001, who captured air samples in two alternating DES, which allowed the analysis with a slow sensor (e.g. up to ~60 s) by switching between the





two reservoirs. The DEC method for NMVOC without any physical pre-sampler (DES) was first implemented by Karl et al., (2002), who used a Proton-transfer-reaction quadrupole mass spectrometer (PTR-QMS) in mass scanning mode similar to a multiplexing technique. This variation was originally coined virtual disjunct eddy covariance (vDEC) (Karl et al., 2002) as no physical device, capturing and storing an air mass, was necessary anymore. The vDEC method is preferable for NMVOC

5    and SVOC that are prone to sampling losses on walls, and is nowadays mostly used for instrumentation that can measure single compounds fast enough (e.g. 0.1 s), so that no physical sample storage is necessary, but where a disjunct method is required to monitor (ie. scan through) multiple compounds. In mass spectrometry this is a particularly attractive method using quadrupole mass spectrometers (QMS) which have to physically scan through a mass spectrum by adjusting internal voltages, so that multiple compounds can be sequentially measured (ie. scanned). To give an example, a Balzers QMG 422

10   QMS needs up to about 0.5 s to internally stabilize voltages and allow the recording of a subsequent compound (ie. molecular ion) (Karl et al., 2002). The sequential mass scanning for 10 molecular ions at 0.1 s sampling rate could therefore require up to 6 s, which would define the DEC interval in this example. Improvements to the high frequency head have shortened internal delay times, but the sequential scanning characteristics of QMS will almost always lead to a vDEC dataset. The covariance (and turbulent flux) between vertical wind and concentration for any DEC dataset can still be

15   calculated, if the wind signal was recorded at the exact time the air sample was taken (e.g. Lenschow et al., 1994):

$$\langle w'c' \rangle = \sum_{i=0}^{N} w_i' c_i' \tag{3}$$

The downside of DES and DEC methods compared to EC is that random errors increase and statistical biases can occur due to undersampling (Lenschow et al., 1994). Another disadvantage of DEC is that co-spectral analysis is not possible due to aliasing, so that high frequency losses due to instrument specific damping for example have to be estimated otherwise.

From an experimental and instrumental point of view three important systematic errors (SE) for EC measurements need to be generally distinguished:

(1)   **Flux averaging:** The total averaging time (T) should capture the entire eddy spectrum contributing to the flux. It
25          has been shown that 30 min averaging intervals are quite suitable for surface layer measurements, and that averaging periods up to 1h can be feasible. Longer averaging periods often suffer from non-stationary conditions (Foken et al., 2010). Averaging periods that are too short will systematically lead to an underestimation of the measured flux (e.g. Massman et al., 2010). Co-spectral analysis can help defining appropriate averaging intervals.

(2)   **Slow sensor response**: A slow sensor will act as a low pass filter, where for example eddies in the inertial
30          subrange cannot be fully resolved anymore. In the surface layer a sensor should ideally be capable of capturing concentration fluctuations at about 10 Hz, but this criterion can be somewhat relaxed depending on the integral time scale ($\tau_F$), and by introducing correction functions to account for damping timescales (e.g. Massman et al., 2010; Wohlfahrt et al., 2009).


(3) **Systematic error due to DEC**: The SE due to disjunct sampling is typically negligible for DEC intervals that are shorter than the integral time scale ($\tau_F$) (e.g. corresponding to the peak in the co-spectrum). The error only increases due to undersampling when the disjunct sampling interval becomes much larger than the $\tau_F$ for a given averaging interval T (Lenschow et al. 1994). In order to keep the SE small for these cases, the sampling interval T will have to increase. For example, $\tau_F$ = 25 s, T = 300 s and a DEC interval of 60 s would lead to a SE of about 23%. Increasing the sampling interval to T = 900 s (1800 s) decreases the SE to 8% (4%). As can be seen from this example, the SE for DEC is mostly negligibly small for surface layer measurements where averaging intervals can be long. Its consideration becomes more significant for airborne measurements (e.g. Karl et al., 2013; Lenschow et al., 1994). It is important to note that gap filling methods (e.g. Spirig et al., 2005) as an alternative to true DEC (Karl et al., 2002), which have been proposed to simplify the data analysis for NMVOC flux measurements, will quickly introduce a SE (Hörtnagl et al., 2010) because these methods act as a low pass filter (discussed above for slow sensors (2)). As an example, if a 10 s DEC interval for $\tau_F$ = 25 s is interpolated by defining an average concentration over the DEC interval, the SE could be as large as 75% as opposed to ~3% due to DEC undersampling.

Random errors (RE) for eddy covariance data-sets have been discussed extensively in the literature (e.g. Lenschow and Kristensen., 1985). Generally, RE are attributed to random uncorrelated measurement noise following Poisson statistics. For averaging intervals much larger than $\tau_F$ the relative errors for DEC and EC scale with $\frac{1}{\sqrt{N}}$; for example if only every 100th sample is recorded due to DEC, its RE will increase by a factor of 10 relative to EC. For trace gases, this square root dependence has been experimentally demonstrated for water vapour by Rinne et al. (2008) and biogenic volatile organic compounds (BVOC) by Turnipseed et al. (2009), who found that DEC intervals for reactive trace gases up to 60 s are feasible for surface layer experiments. Flux detection limits for EC measurements are closely related to RE and have received renewed attention due to the emergence of techniques capable of EC measurements of a wide range of NMVOC (e.g. Karl et al., 2001; Müller et al., 2010; Park et al., 2013) and other reactive trace gases and aerosols (e.g. Sintermann et al., 2011, Held et al., 2007; Nemitz et al., 2008). Measuring fluxes of these trace species is generally more challenging compared to measuring fluxes of heat, $CO_2$ or $H_2O$, because of higher signal-to-noise ratios. An experimental way to determine flux detection limits can be based on the analysis of covariance functions between w (vertical wind) and c (tracer concentration), where the fluctuation far away from the true lag characterizes the flux variance $\sigma_{w'c'}$ (Wienhold et al., 1994). Spirig et al. (2005) proposed to choose a fixed value of the variance $\sigma_{w'c'}$ between 160 to 180 s lag. An alternative way to estimate the error variance ($\sigma_{w'c'}$) is based on the random shuffle method (Billesbach et al., 2011), where one of the time traces (eg. w) is randomly permuted before calculating the covariance between w and c. For both methods the flux detection limit (LOD) can be subsequently defined such that the covariance at lag = 0 must be greater or equal to 3 x $\sigma_{w'c'}$.



A connected topic in this context are procedures for accurate lag-time determinations, which is a much more critical issue for many reactive trace gas flux measurements, because (1) their surface exchange fluxes can be bi-directional and quite low compared to typical flux LODs, and (2) sensor separation is typically a more significant issue than for conventional tower operated trace gas instrumentation (e.g. $CO_2$ and $H_2O$). Karl et al. (2002) have implemented a lag time correction analysis for

BVOC DEC measurements in three steps: (1) Interpolation of the DEC time series to 10 Hz and locating the absolute maximum (or minimum) of the covariance between trace gas concentration (c) and vertical wind (w) within a physically reasonable time window, (2) applying the obtained time shift to the NMVOC DEC dataset; (3) down-sampling high frequency (e.g 10Hz) wind data to the DEC interval and calculating the cross covariance between w and c. Langford et al. (2015) further discussed the issue of lag-time determination for noisy data and devised a recommendation for DEC datasets

that relies on a similar concept. For EC measurements close to flux LOD, Park et al. (2013) suggested to cumulatively add positive covariance functions as a new approach for estimating lag-times at low signal to noise ratios.

Due to the emergence of new analytical instruments capable of EC, DES, or DEC, there is a need to develop customized analysis codes, that can deal with several issues related to accurate data-interpolation, gap-filling, lag-time determination and specification of flux detection limits (Karl et al., 2002; Spirig et al., 2005; Wohlfahrt et al., 2009; Taipale et al., 2010;

Hörtnagl et al., 2010; Langford et al., 2015; Metzger et al. 2017). Here we present and evaluate a unified code, that builds on various improvements reported in the literature (Karl et al., 2001; Karl et al., 2002; Hörtnagl et al., 2010; Taipale et al., 2010; Park et al., 2013; Langford et al., 2015) and streamlines the analysis of EC and DEC datasets for reactive trace gases recorded by various sensors and data acquisition systems. Further, we apply the code to new data on urban EC NMVOC measurements based on a recently developed Proton-transfer reaction (Quadrupole Interface) time of flight mass

spectrometer (PTR-QiTOFMS). The code and routines including a test dataset for aromatic NMVOCs is made available through a data portal.

## 2 Methods

### 2.1 Eddy covariance measurements at the Innsbruck Atmospheric Observatory

An extensive high rate dataset for chemical species was used to develop, test and evaluate the new EC procedure; The

dataset was acquired using a CPEC200 eddy-covariance system by Campbell Scientific®, a PTR-QiTOF mass spectrometer by IONICON Analytik between July and September 2015.

The CPEC200 eddy covariance system was mounted on a tower on the roof of a 10-storey building close to the city center of Innsbruck, Austria (Lat: 47°15'51.66" Lon:11°23'06.82"). A heated inlet line led from the CPEC's sonic anemometer into a laboratory below the roof, where the mass spectrometer was located. The field location is described in more detail by Karl et

al. 2017.



## 2.2 Devices measuring turbulence and concentrations

High time resolution turbulence and concentrations of $H_2O$ and $CO_2$ were measured using a CPEC200 system, which is a closed-path eddy-covariance flux system for monitoring atmosphere-biosphere exchanges of carbon dioxide, water vapor, heat, and momentum. It consists of a closed-path infrared gas analyzer and a 3D sonic anemometer. Data were sampled at 10 Hz.

VOC concentrations were measured using a PTR-QiTOF-MS, a proton transfer reaction time-of-flight mass spectrometer with a quadrupole ion-guide for increased sensitivity (Sulzer et al., 2014). Its capability of measuring concentrations of VOCs (volatile organic compounds) with a sampling rate of 10 Hz makes it suitable for monitoring VOC fluxes using eddy-covariance. For the current dataset the instrument was operated in hydronium mode at standard conditions in the drift tube allowing an E/N of about 112 Townsend. The instrument was set up to sample ambient air from a turbulently purged 3/8" Teflon line. Every seven hours, zero calibrations were performed for 30 minutes providing VOC free air from a continuously purged catalytical converter though a setup of software-controlled solenoid valves. In addition, every other time known quantities of a suite of VOC from a 1ppm calibration gas standard (Apel & Riemer, USA) were added to the VOC free air and dynamically diluted into low ppbv mixing ratios. Typical sensitivities achieved during the experiments were 900 and 1500 counts/ppb for benzene and toluene, respectively.

## 2.3 Eddy Covariance Data Processing

In the following sections the procedure for eddy covariance data processing is described in detail. The procedure is implemented in MATLAB®.

The turbulence data acquired by a sonic anemometer is expected to be available as daily files, i.e. files that contain the data for a whole calendar day. Required variables are the three components of wind speed, $u$, $v$, $w$ and sonic temperature, $T_S$. The files containing concentrations measured by gas analyzers can be of any size. Certain file naming conventions and a simple data format (see appendix) allow for smooth data handling and memory management within the procedure, which automatically loads the files as needed. The input data is separated into user-specified equidistant averaging intervals of typically 30 minutes.

### 2.3.1 Sonic Tilt Correction

A rotation of the wind data is sometimes necessary to correct the tilt angle of the sonic anemometer by aligning its coordinate system's horizontal plane with the average streamlines.

In the present study we observed a dependence of the tilt angle on the respective mean wind direction, so we implemented a **Directional Planar Fit Method** in the innFLUX code. This type of mean wind direction dependent tilt correction allows the user to select site, tower and instrument specific sectors for the analysis. For example, sectors towards the measurement tower, the sonic anemometer back or other instrumentation are often discarded from further analysis. For each wind direction





(in 1-degree steps) the mean wind vectors within a sector of ±15 degrees are taken to calculate the rotational matrix, using all available wind data within that sector passing basic quality criteria. The rotation process for each sector is performed as the planar fit method described by Wilczac et al. (2001). This gives 360 matrices, which are then used for rotating the wind vectors within an averaging interval corresponding to the mean wind direction of that interval. Depending on the size of the

available dataset, the width of the wind sector can be adjusted as necessary (e.g. in case that the ±15-degree sectors do not contain enough data for the planar fit, the angle can be increased).

Alternatively, we provide the option of **Double Rotation** of the wind vectors. This method also works when the amount of wind data is limited, as it determines the rotation angles from individual averaging intervals. The double rotation method is also described by Wilczak et al. (2001).

### 2.3.2 Lag-Time Determination

The accurate determination of lag-times is a particularly important task for a comprehensive analysis of NMVOC and SVOC datasets, where each chemical species might exhibit a slightly different lag-time behavior due to inlet line and instrumental characteristic. Lag-time is the time between the measurement of the wind signal and the concentration of the tracer, which is transported from the inlet near the center of the sonic anemometer to the gas analyzer. An additional time shift can be

introduced due to instrument response time and differences in the internal clock of the recording systems. It has been shown that different methods for lag-time determination can systematically under- or overestimate the flux, see for example Taipale et al. (2010) or Langford et al. (2015).

Usually, the cross-covariance between the fluctuations of the tracer concentration signal and the vertical wind component is determined, assuming that the local maximum (or minimum) corresponds to the lag-time. The maximum (or minimum) of

the covariance has been shown to overestimate the actual absolute value of the flux, because the extremum is likely to be systematically high (or low) due to statistical noise (Taipale et al., 2010). The uncertainty and the magnitude of the over- (or under) estimation tend to increase with decreasing signal-to-noise ratio.

It is therefore suggested to determine the lag-time once for a section of data with high signal-to-noise ratio, and then use this lag-time to time shift the datasets and subsequently infer the covariance at zero lag. Because lag-times can change with time

due to imperfection of the experimental setup, or the fact that they can also vary for different measured tracer species, the assumption of a fixed lag-time tends to underestimate the actual flux at times when the actual lag deviates from the preset lag-time (if calculated only once).

An improvement to this problem was suggested by Taipale et al. (2010) by applying a smoothing filter on the covariance curve prior to determining the lag-time and then taking the location of the maximum (minimum) as the lag-time. This

reduces the influence of statistical noise significantly, and still allows the determined lag-time to follow variations in the actual lag-time.

However, a problem that remains with this approach is that the determination of the extremum in the covariance curve often fails for low signal-to-noise data (e.g. for sections of data where the flux is close to zero), resulting in unreasonable lag-times



and inappropriate flux values determined far off the actual lag-time. To mitigate this problem, we introduce another method, where the covariance curve is accumulated over extended periods during which variations in the lag-time are assumed to be small. This must be ensured by careful experimental setup. Two methods can then be used to determine the lag-time between the tracer and the turbulence signal.

In a first step, the lag-time is determined from the smoothed covariance function by finding the location of the minimum or maximum within a predefined window. Whether to look for the minimum or maximum is determined from the curvature (second derivative) of the strongly smoothed covariance function near the expected lag-time. This is done for each averaging interval.

In a second step, similar to the method applied by Park et al., 2013, the absolute values of all covariance functions over the
full data range (or optionally user-defined sub-periods) are summed up, resulting in a cumulated covariance function with a well pronounced peak, which is typically much smoother than the ones obtained for the individual intervals. The lag-time is then determined from the location of that peak and stored in the results datafile to be applied in following steps. This procedure is conducted separately for every chemical species.

### 2.3.3 Fluxes and Cospectra

Once the lag-time is determined, the flux is calculated from the value of the covariance function at the corresponding lag time. The gas fluxes are then converted to nmol m$^{-2}$ s$^{-1}$ using the ideal gas law.

Optionally the **WPL-correction** as described by Webb et al. (1980) can be applied to the gas fluxes to correct for density effects. When concentrations are measured instead of volume mixing ratios and a water vapor flux is present, the volume
occupied by water vapor would lead to an apparent flux of a tracer in the opposite direction of the water vapor flux. The corrected flux of a tracer with measured concentration $c$ is approximated by:

$$F = \overline{w'c'} + \frac{\bar{c}}{c_d} \overline{w'c_v'},$$
(4)

where $c_v = \bar{c}_v + c_v'$ is the concentration of water vapor, and $c_d$ is the concentration of dry air.

**Cospectra** are calculated as given in the following equation:

$$Co(w',c',f) = \Re[\mathcal{FT}(w')^\star \cdot \mathcal{FT}(c')]$$
(5)

The cospectra are stored as $f \cdot Co(w',c',f)$. Additionally, the cospectra are stored in a scaled dimensionless form as $f \cdot Co(w',c',f)/\text{cov}(w',c')$, with the scaled x-axis being $f \cdot z/U$, where $f$ is the frequency, $z$ is the sensor height and $u$ is the mean wind speed. These scaled cospectra can be averaged and used for spectral correction of the flux results.



### 2.3.4 Quality Control

This chapter describes the determination of quality criteria and how they are best used for filtering results to the desired quality level. Data intervals, for which certain tests fail, are generally not omitted; all results are calculated regardless of the quality checks. It is up to the user to decide how strictly to filter the results based on the output of the quality checks.

Modern sonic anemometers such as that included with the CPEC system produce **diagnostic information** about the status of the system and the data quality. Periods are flagged, when the sonic anemometer does not work reliably (e.g. during heavy rain), or when an error or disturbance is detected. For averaging intervals that contain flagged data, the procedure sets a flag in the output dataset. Intervals with more than a user-defined percentage of flagged or missing data are regarded as unreliable

and omitted by the procedure (results are replaced by NaN – not a number).

**Spike detection and despiking** are commonly applied to the raw data as a first step. Spikes can be the result of instrument issues, e.g. electrical noise, insufficient power supply, water drops between the sonic anemometer transducers. However, it can be difficult to automatically discriminate between instrument issues and physically plausible behavior, so it is

recommended to visually inspect sections of data where spikes are detected, and not remove spikes that are part of the desired signal, thus introducing unintended dampening.

We provide a simple customizable method that can be used for spike detection and flagging. The method is described by Vickers and Mahrt (1997), and detects spikes by comparing their amplitudes to the standard deviation of the time series.

A **steady state test** is implemented to determine if the basic requirements for eddy covariance, namely the steady state condition, is fulfilled. The procedure suggested by Foken and Wichura (1996) is applied, where the averaging interval is divided into short intervals of equal duration (e.g. six intervals of 5 minutes for an averaging interval of 30 minutes). The covariance between the vertical wind component and the property of interest (e.g. a tracer concentration) is calculated for these subintervals and compared to the covariance of the entire averaging interval. It is generally suggested (Foken and

Wichura, 1996) that the covariance of each subinterval should not differ by more than 30% from the covariance of the total interval. While the code outputs all data, the user can specify the steady state threshold during post-processing depending on site specific constraints.

A second requirement for EC that is often used as a quality check is the **test for developed turbulent conditions**, as

described by Foken and Wichura (1996). It is based on flux-variance similarity and makes use of the integral turbulence characteristics of atmospheric turbulence, which depend on stability:

$$\frac{\sigma_x}{X_*} = c_1 \left(\frac{z}{L}\right)^{c_2}, \tag{6}$$





where $\sigma_x$ is the standard deviation of a fluctuating parameter $x$, $X_*$ is the corresponding dynamical parameter (e.g. $\sigma_u$ and $u_*$, or $\sigma_T$ and $T_*$), and $z/L$ is the stability parameter. Table 2 lists the coefficients $c_1$ and $c_2$ for $w$, $u$ and $T$ for different stability ranges as published by Foken et al. (1991, 1996). When $\sigma_x/X_*$ does not differ by more than 50% from the model, using the tabulated values for $c_1$ and $c_2$, developed turbulent conditions are considered to be fulfilled. It is noted that this test is

location dependent, and the parameterization listed in Table 2 can deviate depending on local constraints. We tested the parameterization for Innsbruck and found that with published values for $c_1$ and $c_2$, which were not obtained over urban areas, the ITC test for $<w'T'>$ commonly underestimates turbulent conditions over an urban area.

The **flux detection limit** is estimated by several different criteria:

*Flux noise STD criterion ($LoD_\sigma$)*: Here the values of the covariance function, $F_c(\tau_i)$, at unphysically large lag times, $\tau_i$, are
considered as uncorrelated statistical noise (Wienhold et al., 1994; Spirig et al., 2005). Realizations of $F_c(\tau_i)$ are calculated for $\tau_i$ in intervals of [-180 s, -160 s] and [160 s, 180 s] with $m_F$ and $\sigma_F$ describing the mean value and the standard deviation of $F_c(\tau_i)$. A covariance peak outside of the interval $[m_F - 3\sigma_F, m_F + 3\sigma_F]$ is considered significantly different from the flux noise and thus detectable.

*Flux noise RMSE criterion ($LoD_{RMSE}$)*: A modification of the previous approach was described by Langford et al., 2015,
where instead of the standard deviation, the root of the mean squared deviation of $F_c(\tau_i)$ from zero is calculated.

*Random error criterion*: Finkelstein and Sims, 2001, described an approach based on variance of a covariance between two variables which are first auto- and cross-correlated:

$$\text{RE} = \sqrt{\sum_{t=-m}^{m} f_{w'w'}(t)f_{c'c'}(t) + f_{w'c'}(t)f_{c'w'}(t)}$$

*"Random shuffle" method*: Billesbach (2011) proposed another method for estimating the contribution of random instrument noise. Here one of the two variables (w', c') is randomly time-shuffled before recalculating the covariance,
effectively removing the covariance due to turbulent transport mechanisms, leaving only the random correlations mostly attributable to instrument noise.

*Autocovariance method*: As described by Lenschow (2000), Mauder (2013) or Langford (2015), instrumental noise of the covariance function is estimated by extrapolating the first 4 terms of the autocovariance function to zero lag, and then taking the difference to its value at zero lag.

**3 Results**

**3.1 Tilt Correction**

For the present urban dataset, sonic anemometer tilt correction angles were found to vary significantly with the direction of the mean wind. The observed variation is caused by the topography and the influence of surface roughness surrounding the





urban measurement location, where tall buildings in the vicinity tend to deform the mean streamlines of the airflow from certain directions. For each mean wind direction, the mean wind vectors were therefore split into ±15° intervals to calculate a corresponding tilt correction matrix. The width of the interval is chosen large enough to have sufficient data for the planar fit method, and small enough so that the angle-dependent variation of the correction angles due to the topography is well

resolved. Figure 1 shows the dependency of the sonic tilt correction angles on the mean wind direction, as well as the number of data points contributing to the determination of the tilt correction angles. The first rotation angle α is defined as the pitch angle about the original y-axis, and the second rotation angle β is the roll angle measured about the new x-axis.

Table 1 shows the impact of the two tilt correction methods on the resulting flux. Fluxes were calculated separately without tilt correction, with the double rotation method and with the planar fit method. The results were filtered by basic quality

criteria, and the corrected data were plotted against the uncorrected data. A robust linear fit quantifies the impact of the tilt correction. Applying no tilt correction would tend to overestimate the flux for the current dataset up to approx. 15%.

## 3.2 The Role of Lag-Time Determination

While lag-time determination from a single averaging interval might prove to work well for times where the magnitude of

the flux is sufficiently large, it will fail in many cases where the flux is small (e.g. close to detection limits). This behavior is illustrated in Figure 2, where the individually determined lag-times are plotted against the flux of toluene. For larger flux values the lag-times show little variance, while for smaller fluxes the scatter increases significantly, so that for fluxes close to the detection limit conventional lag-time determination fails, with most lag-times outside a physically meaningful range. The data were chosen for a period of 34 days, when lag-times were constant and variations due to changing instrumental

conditions could be excluded. The dashed line marks the lag-time determined from the cumulated absolute covariance functions over the same data range. Individually determined lag-times for periods with large fluxes show little variability (±0.2 s) about this value. As toluene fluxes decrease the variation of lag-times becomes significant.

This finding suggests that a lag-time from individual averaging intervals can only be determined reliably for periods of large flux (e.g. 3 times above LOD). During periods of small flux and signals just above the limit of detection, it will be very

important to assure that experimental setup produces result in small lag-time variations so that a cumulated covariance function can be used to obtain a reliable lag time determination. A Similar procedure was already described by Park et al. (2013). Our findings suggest that a cumulative lag time determination is preferable for fluxes that are close or below the LOD of individual flux averaging intervals.

## 3.3 Comparison between eddy covariance and disjunct eddy covariance including associated errors

For analytical instruments not allowing to simultaneously measure all chemical species of interest at once, but in a sequential manner, the disjunct eddy covariance (DEC) can be applied. Alternatively, a physical disjunct sampler (Rinne et al., 2001; Warneke et al., 2002) can be used, when the measurement time requires longer integration times (e.g. up to 1 min). Overall





DEC results in a reduction of gross measurement time for each species, as the available measurement time is distributed between individual samples that are spaced apart by the DEC interval (e.g. 10s). Therefore the signal-to-noise ratio is reduced, which makes lag-time determination more difficult and susceptive to statistical errors (see chapter 3.2).

To test the accuracy of lag-time determination with disjunct EC data, artificial DEC datasets were created from a high-resolution EC dataset. Seven different DEC realizations were created by taking every 5[th], 10[th], 20[th], 50[th], 100[th], 200[th] and 500[th] sample of the EC dataset, producing DEC datasets with disjunct sampling intervals between 0.5 and 50 seconds. As an example, Figure 3 shows the DEC vs the EC flux for toluene. As can be seen the scatter around the 1:1 line is largely determined by increased statistical noise due to a smaller amount of data used for the DEC method (every 50[th] sample resulting in a DEC interval of 5s). The slope is close to the 1:1 line indicating that the systematic bias is small.

Systematic errors due to disjunct sampling can be assessed according to Lenschow et al (1994), eq. 55:

$$\frac{F - F(T, \Delta)}{F} = \frac{\Delta}{T} \left\{ \coth\left(\frac{\Delta}{2\tau}\right) - \frac{\frac{\Delta}{T}\left[1 - \exp\left(-\frac{T}{\tau}\right)\right]}{2\sin^2(\Delta/2\tau)} \right\}$$

Here $T$ is the averaging interval, $\Delta$ is the sampling interval, $F$ is the flux and $\tau$ is the integral timescale.

The random error can be estimated according to Lenschow et al (1994), eq. 58:

$$\frac{\sigma_F^2(T, \Delta)}{\mu_f} = \frac{\Delta}{T} \coth\left(\frac{\Delta}{2\tau}\right)$$

Here $\mu_f$ is the variance of the time series with $T \to \infty$.

Figure 4 shows the relative systematic error for the artificial DEC dataset and the model curve according to Lenschow et al. Systematic errors start increasing for DEC intervals larger than 10s and would amount to about a 16% underestimation of the flux for a 300 s DEC interval. In general, we find that the influence of systematic errors is not the limiting constraint when measuring DEC fluxes as long as DEC intervals are lower than about one minute. Figure 5 shows the increase of the random error for the artificially created DEC dataset with increasing sampling interval, and a fit of Lenschow's equation 58 to the data.

**3.4 Flux LOD**

The flux limit of detection (LOD) for high and low signal-to-noise ratios, estimated by four individual methods, are plotted in Figure 6. For practical reasons the first four methods described in section 2.3.4 are implemented in the innFLUX code and are tested using an exemplary 30-minute interval. As can be seen, the random flux (BB) method likely underestimates the flux LOD, which could lead to erroneous flux LODs. The most conservative estimate for a flux LOD is obtained by the *RMSE criterion*. The criterion based on the random error (Finkelstein and Sims, 2001) or standard deviation of the covariance function (Wienhold et al., 1994; Spirig et al., 2005) lie in between of the other two methods.



### 3.5 Spectral Corrections

Spectral corrections can be done by estimating high frequency loss using cospectra. The scaled cospectra provided by the flux routine can be filtered according to desired quality criteria and averaged in order to get a smoother cospectrum than the usually noisy cospectra from the individual averaging intervals.

Here we follow the procedure described by Spirig et al., 2005. The normalized ogives, i.e. the integral of the averaged cospectra of the tracer of interest, are plotted together with the ogives of the sensible heat flux (Fig. 7). From the value of the sensible heat flux' ogive where the tracer's ogive approaches its maximum, we can estimate the high-frequency dampening for $CO_2$ being about 3 percent, and for toluene about 5 percent, respectively.

### 4 Conclusions

We tested the applicability of disjunct eddy covariance and eddy covariance measurements in an urban air matrix using a newly designed software package in MATLAB. The code integrates our current understanding on how to deal with noisy data, which is particularly an issue for emerging high time resolution measurements of a wide range of NMVOCs, SVOCs and other trace gases, that are becoming tractable based on highly sensitive time-of-flight mass spectrometry. We were able to test algorithms for finding cross-covariance peaks based on analyzing distinct isotopes (e.g. the $^{12}$C and $^{13}$C toluene

isotopes). Based on an extensive urban dataset we evaluated realistic LODs for NMVOC flux measurements using a first-generation PTR/SRI-QiTOFMS. For example, for toluene and benzene we found 5-95 percentile LOD ranges of 0.025-0.19 and 0.047-0.49 nmol/(m$^2$s) respectively. The high sensitivity allowed evaluating theoretical expressions of both random and systematic flux errors for NMVOCs due to undersampling. We observe an increase in systematic errors at DEC intervals of about 10s. For DEC intervals of 300s the systematic error amounts to 16%. More important, the increase in random errors at

such long disjunct time intervals becomes large, so that most NMVOC fluxes would likely fall below the observable flux LODs. The presented flux code and analysis also addresses potential sources of errors related to flux measurements above urban canopies. For example, we found that a directional tilt correction improved the accuracy of calculated fluxes by more than 10% for most compounds.

The goal of this work was to develop, test and present a new software package for analyzing EC and DEC data of a wide

range of species, that are relevant for atmospheric chemistry and related disciplines. The code along with a test dataset will be made available through a Git repository. While we focused particularly on the applicability towards urban flux measurements in this study, this software package is expected to be applicable in other environments that are not horizontally homogeneous.



## Appendix A

### A.1 Using the Flux Routine

The flux routine is written in MATLAB (release R2018a). The input files need to be prepared as MATLAB data files as described in the section below.

If sonic tilt correction using the planar fit method should be applied, a tilt correction file must be created first. This can be done by the **innFLUX_step0** routine. Choose a dataset as large as possible, containing sonic data of a period during which the sonic anemometer was not moved. The routine will create a file 'tilt_correction.mat', which can be used as input in the following routine.

The actual flux routine is divided in two steps. The **innFLUX_step1** routine calculates all meteorological data and the fluxes
using the lag-time determined from each single averaging interval. A file 'results.mat' is created. The lag-times in this file should be checked before proceeding with the second step of the flux routine. If the lag-times scatter around the same mean value during the whole measurement period, no further action is required before proceeding with the next step. If the lag-time shows sudden changes, the data range should be divided into several subranges, during which the lag-time scatters around the same mean value. This is done by adjusting the *data_segments* parameter in the parameter file. The
**innFLUX_step2** routine calculates the lag-times from covariances cumulated over the full data range or subranges of consistent lag-times, if *data_segments* are defined.

### A.2 Input Data Format

This section describes the format of the input data and the parameters in the parameter file.
All input files are MATLAB data files (.mat), unless otherwise stated.

### A.2.1 Parameter File

The parameter file is named 'innFLUX_parameters.mat' and contains path definitions and several parameters for configuring the flux routine.

- *output_folder:* folder where the output files will be stored
- *sonic_files_folders{}:* one or more folders containing the sonic data files
- *tracer_files_folders{}:* one or more folders containing the tracer files
- *tracer_files_prefix:* filename prefix of the tracer files
- *tracer_files{}:* one or more tracer files, only used if *tracer_files_folders{}* is empty
- *tilt_correction_filepath:* path of a tilt correction file created by the tilt correction routine
- *pressure_filepath:* path of an optional file containing pressure data
- *irga_columns[]:* column indices of IRGA data; leave empty if no IRGA data is present
- *irga_names{}:* names of IRGA data; leave empty if no IRGA data is present





- *irga_H2O_index:* determines which of the IRGA tracers is H2O; needed for temperature and WPL correction
- *irga_flag_column:* column index of the IRGA data quality flag
- *tracer_indices[]:* indices of tracers to be processed; if empty, all tracers are processed
- *data_segments[]:* data segments, for which a global lag-time is used each; if empty, the whole data period is used; format: vector of timestamps defining the borders of the segments, *e.*g. for 2 segments: data_segments = [datenum(2019,7,1) datenum(2019,7,21) datenum(2019,8,7)];
- *SONIC_ORIENTATION:* orientation of the sonic anemometer in degrees relative to its local coordinate system
- *SENSOR_HEIGHT:* sensor height in meters above the roughness height
- *WINDOW_LENGTH:* length of the averaging window in samples
- *SAMPLING_RATE_SONIC:* sonic anemometer sampling rate in samples per second
- *SAMPLING_RATE_TRACER:* trace gas analyzer sampling rate in samples per second
- *DISJUNCT_EC:* if 1, disjunct eddy covariance is applied
- *DETREND_TRACER_SIGNAL:* if 1, tracer signal detrending is applied instead of Reynold's averaging
- *MAX_LAG:* maximum lag when calculating covariances, in samples
- *LAG_SEARCH_RANGE:* range (+/-) for lag-time search, in samples
- *COVPEAK_FILTER_LENGTH:* smoothing length of filter applied to covariance function prior to finding the covariance peak, in samples
- *NUM_FREQ_BINS:* number of logarithmically spaced frequency bins for cospectra
- *FREQ_BIN_MIN:* lowest frequency bin for scaled cospectra
- *FREQ_BIN_MAX:* highest frequency bin for scaled cospectra
- *TILT_CORRECTION_MODE:* 0: no tilt correction, 1: double rotation, 2: directional planar fit
- *APPLY_WPL_CORRECTION:* if 1, WPL correction is applied
- *COMPLETENESS_THRESHOLD:* threshold value for completeness of input data, 0.0 - 1.0; below this threshold, results are not calculated and filled with NaN (not a number)
- *DEFAULT_PRESSURE:* fixed pressure used if no pressure is given in the input files; in hPa / mbar

**A.2.2 Sonic Data File Format**

Sonic files must contain exactly one day's data each and be named 'wyyyymmdd.mat', where yyyy, mm and dd stands for the corresponding year, month, and day, respectively, e.g. 'w20190725.mat'. The routine can find all files following this naming convention inside one or more given folders. The sonic data files must contain the data of a full day each. Missing data should be filled with NaN (not a number) as a placeholder. Sonic files consist of a matrix, where the columns contain the following data:

Column 1: MATLAB timestamp





Column 2: reserved

Column 3: first horizontal component of the wind vector (in the sonic anemometer's coordinate system)

Column 4: second horizontal component of the wind vector

Column 5: vertical component of the wind vector

Column 6: sonic virtual temperature

Column 7: sonic flag; this is normally 0, and it is different from 0 when there was an error in wind data acquisition

Column 8..N (optional): concentration of a tracer measured by infrared gas analyzer (IRGA)

Column N+1 (optional): IRGA flag; normally 0, and different from 0 when there was an error in IRGA data acquisition

If one of the IRGA tracers is water vapor, the unit is expected to be permille.

**A.2.3 Tracer Data File Format**

The tracer data can be provided as files containing one day's data each named with the prefix *tracer_files_prefix* as given in the parameter file followed by the corresponding day's date in the format 'yyyymmdd', e.g. 'ptrms20190725.mat'. The routine can find all files following this naming convention inside one or more given folders (*tracer_files_folders*).

Alternatively, the tracer data can be provided in one or more files of arbitrary name containing an arbitrarily large amount of 15 data (*tracer_files*).

The tracer data files must consist of a structure containing the fields *header* and *data*. The *header* field is a cell array of strings containing the names of the columns in the *data* field. The first column in the *data* field contains the MATLAB timestamps, the other columns contain the tracer concentration data in ppb.

**A.2.4 Pressure Data File Format**

Ambient pressure data can be provided in a separate MATLAB data file consisting of a struct containing the fields *time* and *p*. The field *time* must contain MATLAB timestamps, and the field *p* must contain the corresponding pressure data in hPa (mbar). Here pressure can be provided at lower sampling rates and is interpolated by the flux routine as needed.

**A.3 Output Data**

**A.3.1 Results File**

The main output file is a MATLAB data file named 'results.mat'. It consists of a struct containing the following fields:

- *time:* timestamp of the beginning of each averaging interval, in MATLAB time
- *hour:* hour of day
- *freq:* frequency axis of cospectra, in 1/s
- *freq_scaled:* frequency axis of scaled cospectra, f·z/u, dimensionless
- *MET.uvw:* mean wind speed vector with u component pointing into direction of mean wind, in m/s





- *MET.std_uvw:* standard deviation of wind speed components u, v, w, in m/s
- *MET.hws:* horizontal wind speed, in m/s
- *MET.wdir:* horizontal wind speed, in m/s
- *MET.tilt.P:* tilt correction matrix applied to wind vectors
- *MET.uw:* covariance of along-wind and vertical wind component, $\langle u'w' \rangle$, in $m^2/s^2$
- *MET.vw:* covariance of cross-wind and vertical wind component, $\langle v'w' \rangle$, in $m^2/s^2$
- *MET.uv:* covariance of along-wind and cross-wind component, $\langle u'v' \rangle$, in $m^2/s^2$
- *MET.uu:* auto-covariance of along-wind component, $\langle u'u' \rangle$, in $m^2/s^2$
- *MET.vv:* auto-covariance of across-wind component, $\langle v'v' \rangle$, in $m^2/s^2$
- *MET.ww:* auto-covariance of vertical wind component, $\langle w'w' \rangle$, in $m^2/s^2$
- *MET.ust:* friction velocity, u*, in m/s
- *MET.T:* temperature, in Kelvin
- *MET.std_T:* standard deviation of mean Temperature, in Kelvin
- *MET.wT:* temperature flux, $\langle w'T' \rangle$, in K·m/s
- *MET.L:* Obukhov length, in m
- *MET.zoL:* stability Parameter, z/L, dimensionless
- *MET.cospec_wT:* cospectrum for wT, f·Co(w',T',f)
- *MET.cospec_wT_scaled:* scaled cospectrum for wT, f·Co(w',T',f)/cov(w',T')
- *MET.p:* pressure, in hPa
- *MET.theta:* potential temperature, in Kelvin
- *MET.theta_v:* virtual potential temperature, in Kelvin
- *MET.wtheta:* potential temperature (heat) flux, $\langle w'theta' \rangle$, in K·m/s
- *MET.wtheta_v:* virtual potential temperature (buoyancy) flux, $\langle w'\Theta_v' \rangle$, in K·m/s
- *MET.qaqc.completeness:* fraction of sonic data used in this averaging interval
- *MET.qaqc.SST_wT:* steady state test for wT, relative deviation
- *MET.qaqc.ITC_w:* relative model deviation of integral turbulence characteristics test for w
- *MET.qaqc.ITC_u:* relative model deviation of integral turbulence characteristics test for u
- *MET.qaqc.ITC_T:* relative model deviation of integral turbulence characteristics test for T
- *MET.qaqc.cospec_wT_integral:* integral of cospectrum for wT
- *IRGA(i).name:* name of i[th] tracer measured by IRGA
- *IRGA(i).mean:* mean concentration
- *IRGA(i).std:* standard deviation of concentration





- *IRGA(i).flux:* flux, in nmol/m$^2$/s
- *IRGA(i).cospec:* cospectrum for w and tracer concentration, f·Co(w',c',f)
- *IRGA(i).cospec_scaled:* scaled cospectrum for w and tracer concentration, f·Co(w',c',f)/cov(w',c')
- *IRGA(i).qaqc.completeness:* fraction of tracer data used in this interval
- *IRGA(i).qaqc.SST:* steady state test for tracer, relative deviation
- *IRGA(i).qaqc.flux_SNR:* flux signal-to-noise ratio
- *IRGA(i).qaqc.flux_noise_std:* standard deviation of flux noise far off integral time scale
- *IRGA(i).qaqc.flux_noise_mean:* mean flux noise far off integral time scale
- *IRGA(i).qaqc.flux_noise_rmse:* RMSE of flux noise far off integral time scale
- *IRGA(i).qaqc.random_error_FS:* random error as described by Finkelstein and Sims 2001
- *IRGA(i).qaqc.random_error_noise:* random error noise estimated according to Mauder 2013
- *IRGA(i).qaqc.random_flux:* random flux level estimated by random shuffle criterium (Billesbach 2011)
- *IRGA(i).qaqc.cospec_integral:* integral of cospectrum
- *TRACER(i).name:* name of i[th] tracer
- *TRACER(i).mean:* mean concentration, in ppb
- *TRACER(i).std:* standard deviation of (detrended) tracer concentration, in ppb
- *TRACER(i).lagtime1:* lag-time determined individually for this averaging interval, in seconds
- *TRACER(i).flux1:* flux calculated using lagtime1, in nmol/m$^2$/s
- *TRACER(i).wpl_corr:* WPL correction term, in nmol/m$^2$/s
- *TRACER(i).cospec:* cospectrum for w and tracer concentration, f·Co(w',c',f)
- *TRACER(i).cospec_scaled:* scaled cospectrum for w and tracer concentration, f·Co(w',c',f)/cov(w',c')
- *TRACER (i).qaqc.completeness:* fraction of tracer data used in this interval
- *TRACER (i).qaqc.SST:* steady state test for tracer, relative deviation
- *TRACER (i).qaqc.flux_SNR:* flux signal-to-noise ratio
- *TRACER (i).qaqc.flux_noise_std:* standard deviation of flux noise far off integral time scale
- *TRACER (i).qaqc.flux_noise_mean:* mean flux noise far off integral time scale
- *TRACER (i).qaqc.flux_noise_rmse:* RMSE of flux noise far off integral time scale
- *TRACER (i).qaqc.random_error_FS:* random error as described by Finkelstein and Sims 2001
- *TRACER (i).qaqc.random_error_noise:* random error noise estimated according to Mauder 2013
- *TRACER (i).qaqc.random_flux:* random flux level estimated by random shuffle criterium (Billesbach 2011)
- *TRACER (i).qaqc.cospec_integral:* integral of cospectrum





**Data and code availability**

Measurement data for relevant trace gases are available upon request from the corresponding author. The analysis code is available as supplement.

**Author contributions**

MS, MG and TK conceived the data analysis codes and study. MG and MS conducted the measurements for the study. MS, MG and TK analyzed the data, interpreted the results and wrote the paper. TM contributed to editing the manuscript.

**Competing interests**

The authors declare that they have no conflict of interest.

**Acknowledgment**

This work was supported by the Austrian National Science fund (FWF) through grant P30600.

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





**Table 1: Comparison of calculated fluxes using two different anemometer tilt correction methods with respect to no tilt correction. Slopes are determined by a robust linear fit through corrected data vs. uncorrected data, and include 95% confidence bounds.**

|  | Slope of double rotation vs. no rotation | Slope of planar fit vs. no rotation |
|---|---|---|
| $\langle w'T'\rangle$ | 1.085 ±0.022 | 0.861 ±0.027 |
| $CO_2$ flux | 0.964 ±0.023 | 0.897 ±0.027 |
| Toluene flux | 0.974 ±0.025 | 0.854 ±0.024 |

5  **Table 2: Dependence of the integral turbulence characteristics for wind components w, u and temperature T on the stratification (Foken and Wichura 1996, Foken et al., 1991).**

| $z/L$ | $\sigma_w/u_*$ | $\sigma_u/u_*$ | $\sigma_T/T_*$ |
|---|---|---|---|
| $z/L < -1$ | $2.00(-z/L)^{1/6}$ | $2.83(-z/L)^{1/6}$ | $1.00(-z/L)^{-1/3}$ |
| $-1 < z/L < -0.0625$ | $2.00(-z/L)^{1/8}$ | $2.83(-z/L)^{1/8}$ | $1.00(-z/L)^{-1/4}$ |
| $-0.0625 < z/L < 0$ | 1.41 | 1.99 | $0.50(-z/L)^{-1/2}$ |





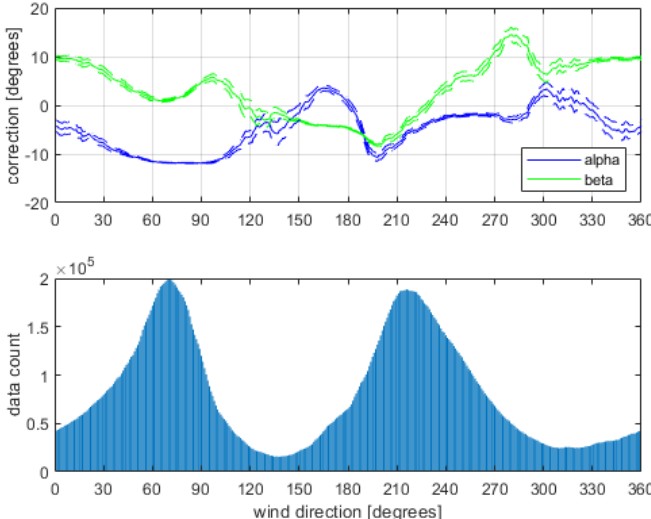

**Figure 1: Top: sonic anemometer tilt correction angles in dependence of the mean wind direction. For each mean wind direction, all available half-hour intervals of data with mean wind direction within +/-15 degrees were taken for determining the tilt correction angles according to the planar fit method. The dashed lines show 95% confidence bounds estimated by bootstrapping. Bottom: The histogram shows the number of half-hour intervals contributing to the calculation of the correction angles.**

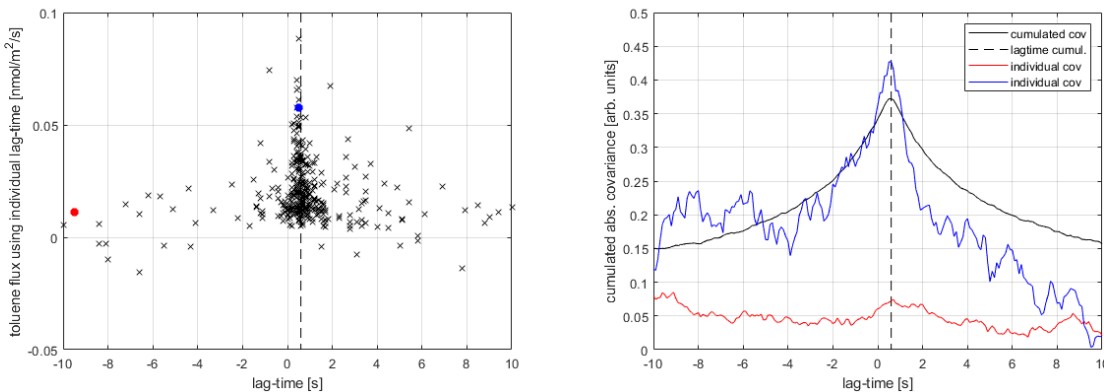

**Figure 2: Left: lag-time determined from individual averaging intervals in dependence of the flux. For periods of small flux individually determined lag-time shows large scatter. Right: individual and cumulated abs. cross-covariance function for toluene $^{13}C$ isotope used for estimating the lag-time. The dashed line shows the estimated lag-time. The lag-times corresponding to the individual covariance functions are shown as red and blue dots in the graph on the left.**



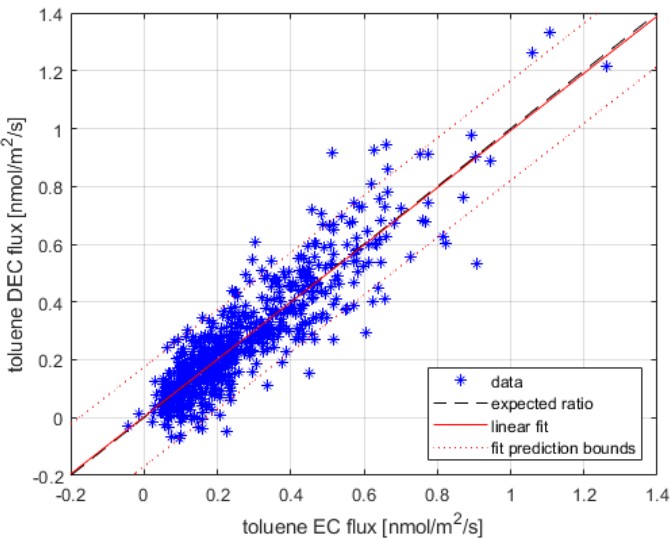

**Figure 3:** **Toluene flux determined by DEC (taking every 50th sample) vs. toluene flux determined by EC. The fitted curve shows very good correspondence of the DEC and EC flux results, with slope close to 1. Slope: 0.99, offset: 0.003, $R^2$: 0.89**

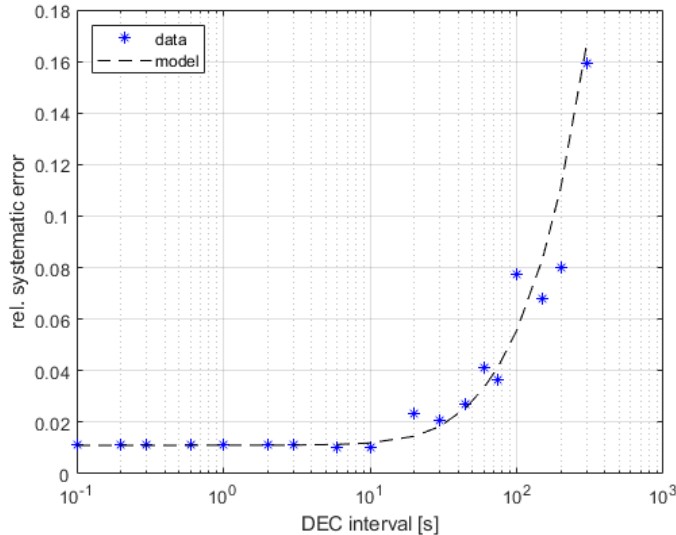

**Figure 4: Systematic error of toluene flux with increasing DEC sampling interval Δt determined by artificially created DEC datasets and according to Lenschow et al. 1994**





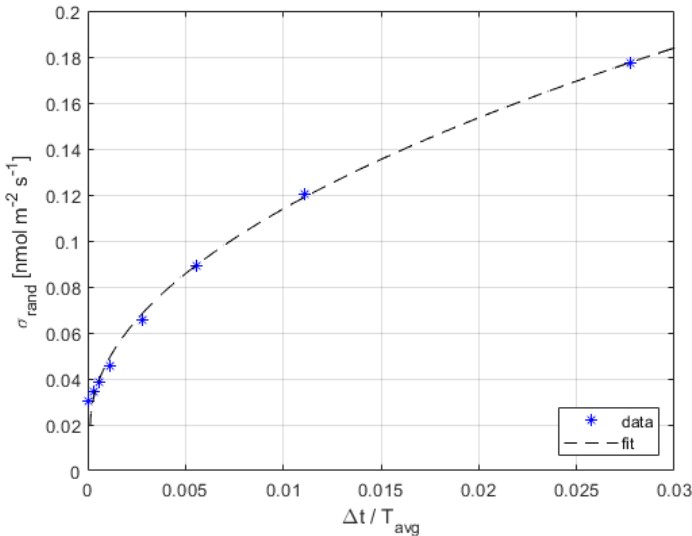

**Figure 5: Random error of toluene flux with increasing DEC sampling interval Δt determined by artificially created DEC datasets following a square root relation.**

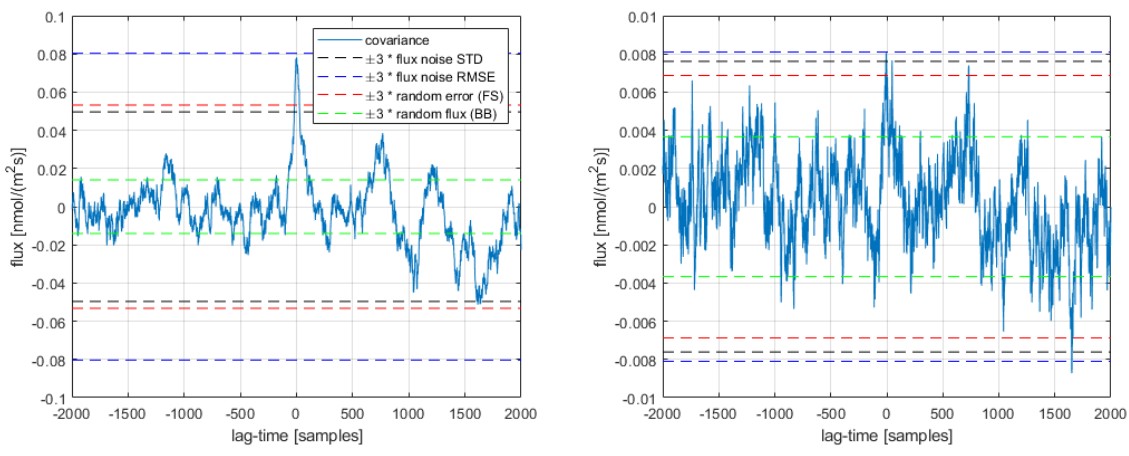

**Figure 6: Covariance function and limit of detection (LOD) for Toluene (left panel) and its $^{13}$C isotope (right panel) estimated using different approaches.**





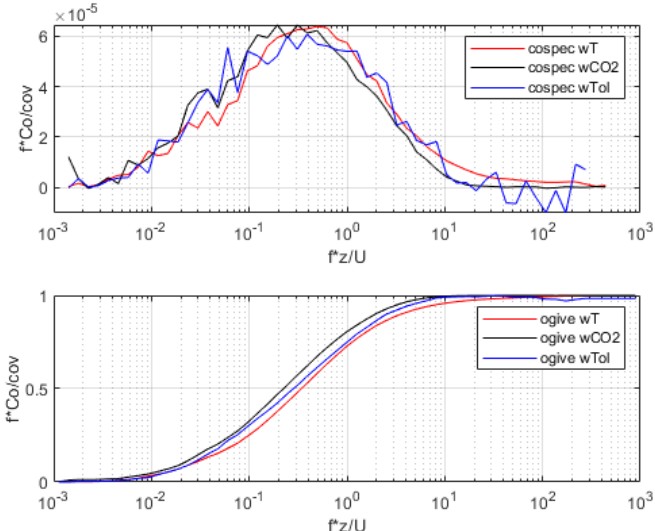

**Figure 7: Average scaled cospectra and corresponding ogives for heat flux, CO2 and toluene.**