# Peer review of "innFLUX – An open source code for conventional and disjunct eddy covariance analysis of trace gas measurements: An urban test case"

_Atmospheric Measurement Techniques, 2019_

## Referee Comment (RC1) · Janne Rinne (Referee) · 4 Oct 2019

Dear Striednig et al.,

Development of new software to serve the community working on VOC flux measurement is recommended activity. This far everyone working on the field has been developing their own codes, which takes resources and possibly leads to systematic differences in processing of their data. Thus, I see a lot of value on this manuscript. I have a few comments stated below.

1. As the new software is able to process also conventional eddy covariance data, I would like to see some comparison between data processed with the new software and established EC post-processing softwares, e.g. EddyPro, EddyUH (Mammarella et al., 2016).

2. Test site is challenging from micrometeorological point of view. How about including data from some more ideal measurement site to analysis?

3. The equation (1) is not correct and I am not satisfied with how Eq. (2) is derived.

The correct form of the conservation equation for a scalar s can be written in the form

$$\frac{\partial \bar{s}}{\partial t} = -\bar{u}\frac{\partial \bar{s}}{\partial x} - \bar{v}\frac{\partial \bar{s}}{\partial y} - \bar{w}\frac{\partial \bar{s}}{\partial z} - \frac{\partial \overline{(u's')}}{\partial x} - \frac{\partial \overline{(v's')}}{\partial y} - \frac{\partial \overline{(w's')}}{\partial z} + D\frac{\partial^2 \bar{s}}{\partial x^2} + D\frac{\partial^2 \bar{s}}{\partial y^2} + D\frac{\partial^2 \bar{s}}{\partial z^2} + Q,$$

where $D$ is the molecular diffusivity and $Q$ is the chemical source/sink term (e.g. Stull, 1988). Assuming horizontal homogeneity all terms with horizontal derivatives will disappear. Horizontal homogeneity at flat surface also leads to vertical wind speed $w$ to be zero, as at the surface $w$ is zero and

$$\frac{\partial u}{\partial x} + \frac{\partial v}{\partial y} + \frac{\partial w}{\partial z} = 0,$$

where horizontal derivatives are zero. Adding stationarity assumption, the time derivative also disappears. Further, if the chemical lifetime of the trace gas in question is much longer than turbulent mixing time scale (Rinne et al. 2012), the source term $Q$ vanishes as well. Thus, we are left with two terms we can integrate from surface ($z=0$) to measurement height ($z=h$),

$$\int_0^h \frac{\partial \overline{(w's')}}{\partial z} dz = \int_0^h D\frac{\partial^2 \bar{s}}{\partial z^2} dz,$$

leading to

$$\left(\overline{w's'}\right)_h - \left(\overline{w's'}\right)_0 = D\left(\frac{\partial \bar{s}}{\partial z}\right)_h - D\left(\frac{\partial \bar{s}}{\partial z}\right)_0.$$

Noticing that turbulent flux, $\overline{w's'}$, is orders of magnitudes higher than diffusive flux at typical flux measurement height (1-30 m), and that the turbulent flux at the surface goes to zero as the vertical movements go to zero, we are left with

$$\left(\overline{w's'}\right)_h = -D\left(\frac{\partial \bar{s}}{\partial z}\right)_0,$$

i.e. turbulent flux at the measurement height *h* equals the diffusive surface flux, which we are usually interested in. There are different formulations of this, by e.g. expressing biological sources as term Q, but they will lead to similar final result in which the turbulent flux equals the sources below the measurement level.

4. The required response time of sensor for eddy covariance is stated in the manuscript to be on the order of 0.1 s. However, e.g. Rantala et al. (2014) have shown the response time of a quadrupole PTR-MS to be around 1.2 s. Furthermore, they showed that above forest this response time lead to flux underestimation ranging from below 10% in daytime, to about 20% during night. Thus, if the response time of the instrument is in that range, sampling output at higher frequency does not actually lead to better frequency response. The response time of PTR-ToFMS used in the measurements to test the software is not stated in the paper. The statement on page 3, lines 5-6 "…is nowadays mostly used for instrumentation that can measure single compounds fast enough (e.g. 0.1 s)…" may be bit optimistic.

5. On page page 4, lines 4-6 the authors give an example of systematic error caused by disjunct sampling as 23%, with sampling interval Δ=60 s, integral time scale τ=25 s, and flux averaging time T=300 s, based on equation by Lenschow et al. (1994). However, setting Δ=0.1 s, i.e. typical conventional eddy covariance sampling frequency, leads to flux underestimation of 15%. Thus, more than half of the flux underestimation is not due to disjunct sampling, but rather undersampling the low-frequency contribution by this very short (5 min) flux averaging period. Similarly, for other flux averaging periods shown most of the underestimation does not derive from disjunct sampling. Furthermore, for typical surface flux measurement averaging periods (30 min = 1800 s), the underestimation with sampling interval of even as long as 3 min causes flux underestimation of less than 10%. The authors give a similarly misleading statement on page 12, lines 15-16.

6. Sensor separation (Page 5, line 3) is usually smaller source to lag time than is the long sample tube in the case of closed path analyzers such as PTR-MS.

7. I got the impression that the software is not performing frequency corrections to fluxes using cospectral densities. One could also correct for high-frequency losses in DEC measurements, if system response time is known, e.g. by test-run by the same system with continuous sampling.

Minor comments

Page 6, line 10: That was the flow rate in the 3/8" sample line?

Page 6, line15: Instrument response time, in the sense of the 1-order system, does not cause significant time shift.

Technical comments

Please check that the chemical compounds are properly expressed with subscripts ($CO_2$, $CH_4$, $H_2O$ instead of CO2, CH4, H2O).

Page 4, line 26: "…higher signal-to-noise ratio…" should be "…lower signal-to-noise ratio…"

Page 6, line 3: Typo, Wilczac should be Wilczak.

References

Mammarella, I., Peltola, O., Nordbo, A., Järvi, L., Rannik, Ü., Quantifying the uncertainty of eddy covariance fluxes due to the use of different software packages and combinations of processing steps in two contrasting ecosystems. Atmos. Meas. Tech., 9, 4915–4933, 2016.

Rantala, P., R. Taipale, J. Aalto, M.K. Kajos, J. Patokoski, T.M. Ruuskanen & J. Rinne: Continuous flux measurements of VOCs using PTR-MS - reliability and feasibility of disjunct eddy covariance, surface layer gradient, and surface layer profile methods. Boreal Environment Research, 19 (suppl. B), 87-107, 2014.

Rinne, J., T. Markkanen, T.M. Ruuskanen, T. Petäjä, P. Keronen, M.J. Tang, J.N. Crowley, Ü. Rannik & T. Vesala: Effect of chemical degradation on fluxes of reactive compounds – a study with a stochastic Lagrangian transport model. Atmospheric Chemistry and Physics, 12, 4843–4854, 2012.

Janne Rinne
Lund University

---

## Short Comment (SC1) · 31 Oct 2019

This is a short and well written manuscript introducing a procedure for processing turbulent fluxes of trace gases by disjunct eddy covariance over relatively heterogenous surfaces. The disjunct eddy covariance method has the ability of measuring fluxes using instrumentation not capable of collecting data at very high frequency (e.g. 10-20 Hz) as requested by the true eddy covariance method.

Although the method has been previously described and tested over urban surfaces, this manuscript summarizes quite well the steps and corrections needed for its application. After addressing the following comments, the manuscript can be considered for

publication.

Main comments

- 'Unified' may not be an appropriate term for the title. This reviewer agrees in general with the assumptions and corrections included in the proposed methodology, but not all researchers may do it. Some debate exists on how to postprocess turbulent fluxes.

- A flowchart will help to visualize the order of the steps for postprocessing disjunct fluxes.

- The introduction should explain the need for measuring fluxes by eddy covariance over urban surfaces, particularly of speciated VOCs.

- Velasco et al. (2005, doi:10.1029/2005GL023356) deployed by first time a PTR-MS for measuring turbulent fluxes over an urban surface using the disjunct eddy covariance method. Some of the corrections and assumptions discussed here were also discussed by them.

Specific comments Page (lines)

1(8-9) ". . . and disjunct eddy covariance flux data."

1(9) Define acronyms every time they are used by first time. Please check this throughout the whole text. Many acronyms were used without being properly defined.

1(16) What about the met data necessary to compute turbulent fluxes?

1(24) Replace "surface fluxes" by "turbulent fluxes".

2(6) Use italic fonts for referring to variables. Check this throughout the whole text.

2(7) in the horizontal dimension?

2(11-12) fast and highly accurate sensors.

2(12-14) Consider that the atmospheric reactivity of some species limits the application

of the eddy covariance method for measuring turbulent fluxes. Some species react faster than the time taken by the air sample to reach the height of the instrumented tower. This is a particular constraint in polluted urban atmospheres.

3(18) Why is the co-spectral analysis important? What does it show?

3(24-26) The averaging time peirod depends also on the roughness elements' height. For flux measurements over smooth surfaces such as lakes and grasslands, for example, averaging time periods of 10-15 min are used, while for measurements over tall canopies in forested and urban environments, averaging periods of 30 min are common.

3(29-30) Define inertial subrange.

4(23-24) Explain how you reached this figure.

5(27-30) You could save the reader of searching in a second article to learn about the eddy covariance flux system used here as a test case. Provide at least the local climate zone, land cover, measurement height, mean roughness elements height and zero-plane displacement height.

6(13) How many VOC species and of which groups (i.e. olefins, aromatics, etc.)?

6(20) What about data from a low frequency met sensor for flux corrections. The sonic/virtual temperature is not the absolute temperature.

8(19) In most urban environments moisture in the air is inherent.

8(25) But a co-spectra analysis is not feasible for DEC as explained above .....

---

## Author Comment (AC1) · 20 Dec 2019

**Comment 1:** As the new software is able to process also conventional eddy covariance data, I would like to see some comparison between data processed with the new software and established EC post-processing softwares, e.g. EddyPro, EddyUH (Mammarella et al., 2016).

**Reply:** As suggested we have included a comparison using EddyPro. The software codes yield comparable results. The regressions for wT and wCO2 fluxes, for example, show $R^2$s of 0.99 and 0.97, respectively, slopes are 0.95 and 1.02, respectively.

**Changes:** The revised manuscript now also includes the suggested comparison, which was included in the supplementary files (Chapter S2 and Figures S2 und S3).

**Comment 2:** Test site is challenging from micrometeorological point of view. How about including data from some more ideal measurement site to analysis?

**Reply:** We share the notion that flux measurements in urban areas might appear more complex at first sight, but would like to point out that most towers in areas exhibiting high reactive gas fluxes have site specific challenges (Foken et al., 2012; , Park et al., 2013, Rantala et al., 2016:). The definition of an ideal measurement site is therefore often difficult. For an urban location we would argue that the site is of low to intermediate complexity (ie. homogeneous footprint in the two main flux footprint corridors, flat terrain (valley bottom)).The advantage of the present dataset is that (1) we deal with well characterized anthropogenic tracers (e.g. aromatic compounds), and (2) the PTR-qTOF-MS instrument possesses enough sensitivity to capture carbon isotope fluxes (e.g. 13C Toluene). We see this as an essential added benefit to the evaluation presented here.

**Comment 3:** The equation (1) is not correct and I am not satisfied with how Eq. (2)is derived.

**Reply:** The equation already included some simplifications (ie <w>~0) as pointed out by the reviewer.

**Changes:** We modified the derivation as and included the complete derivation as suggested by Rinne.

**Comment 4:** The required response time of sensor for eddy covariance is stated in the manuscript to be on the order of 0.1 s. However, e.g. Rantala et al. (2014) have shown the response time of a quadrupole PTR-MS to be around 1.2 s. Furthermore, they showed that above forest this response time lead to flux underestimation ranging from below 10% in daytime, to about 20% during night. Thus, if the response time of the instrument is in that range, sampling output at higher frequency does not actually lead to better frequency

response. The response time of PTR-ToFMS used in the measurements to test the software is not stated in the paper. The statement on page 3, lines 5-6 "...is nowadays mostly used for instrumentation that can measure single compounds fast enough (e.g. 0.1 s)..."may be bit optimistic.

*Reply:* Thank you for this valuable comment - we fully agree with the reviewer that with regards to cospectral attenuation the response time of a sensor (rather than the output frequency) is one of the key parameters that may govern the high frequency loss of an eddy covariance system. This holds particularly for the data of the VOC EC system exemplarily used here for the demonstration of the capabilities of the innFLUX open source code. The response time of a PTR-MS system depends in first order on the exchange rate of the sample gas in the inner volume of the instrument, which is largely governed by the drift tube volume and the volumetric flow, and for sticky compounds, on the properties and area of wetted surfaces. In both regards small dead volumes, reduced surface areas, inert materials, low pressure and high temperature improve the response time. Since the introduction of the PTR-MS method in the 1990s the achievable response times had increasingly improved from ~0.8 s in 1999 (Karl et al 2001) and about 1 s in 2000 (Rinne et al 2001) to at least 0.1 s in 2001 (Karl et al. 2002) as a consequence of design improvements driven by the requirements of the EC technique. The PTR-Qi-TOF instrument here has a characteristic time constant of 0.08 s. It is indeed important to point out how the response time of a closed path analyzer (amongst other dampening effects) affects the cospectral attenuation and how the EC flux can be corrected for such high frequency losses.

*Changes:* We conducted a thorough analysis of the cospectral behavior of the VOC EC system described here based on Foken et al. (2012a) and Lee et al. (2004). The added Chapter S4 in the supplement now guides the reader how to derive a model cospectrum from quality checked individual half-hour cospectra (example in Figure S3). It shows how to determine transfer functions describing high frequency lossed due to sensor separation, sonic path averaging, sensor path averaging (PTR-MS response) and tube attenuation, and how these attenuations cause loss of cospectral density at high frequencies, thus underestimating the flux (Figure 7). The new Chapter 3.5 in the main text points out the cospectral information calculated and stored by innFLUX, mentions both the experimental approach and the theoretical approach for the correction of high frequency losses, gives the user guidance which approach might be more appropriate, and details the procedure (reference to Chapter S4 and Figure S3) and results (Figure 7 and Figure S4) of the cospectral analysis.

*Comment 5:* On page page 4, lines 4-6 the authors give an example of systematic error caused by disjunct sampling as 23%, with sampling interval D=60 s, integral time scale t=25 s, and flux averaging time T=300 s, based on equation by Lenschow et al. (1994).However, setting D=0.1s, i.e. typical conventional eddy covariance sampling frequency, leads to flux underestimation of 15%. Thus, more than half of the flux underestimation is not due to

disjunct sampling, but rather undersampling the low-frequency contribution by this very short (5 min) flux averaging period. Similarly, for other flux averaging periods shown most of the underestimation does not derive from disjunct sampling. Furthermore, for typical surface flux measurement averaging periods (30 min = 1800s), the underestimation with sampling interval of even as long as 3 min causes flux underestimation of less than 10%.The authors give a similarly misleading statement on page 12, lines 15-16.

*Reply:* We acknowledge that the numbers in the original manuscript represent the sum of systematic errors due to DEC and block averaging interval. Figure 4 shows that DEC intervals of about 100s for T=1800 s would lead to a 10% bias. The general conclusion is that at certain DEC intervals the block averaging interval needs to increase and problems due to non-stationarity might arise, when T becomes too large.

*Changes:* We clarified the issue of DEC errors and averaging periods in the revised manuscript

*Comment 6:* Sensor separation (Page 5, line 3) is usually smaller source to lag time than is the long sample tube in the case of closed path analyzers such as PTR-MS.

*Reply:* OK we added that a long sample tube is also a relevant parameter.

*Comment 7.* I got the impression that the software is not performing frequency corrections to fluxes using cospectral densities. One could also correct for high-frequency losses in DEC measurements, if system response time is known, e.g. by test-run by the same system with continuous sampling.

*Reply:* We acknowledge that high frequency losses can be accounted for by using empirical formulas (e.g. Horst et al., 1997) and adjusting these to real 10 Hz data. The code outputs all spectral information, so that the user can determine system response times from co-spectral comparison. The code is not automatically implementing a correction to the data, since we believe these things are instrument specific and really need to undergo thorough evaluation by the experimentalist. Depending on the nature of a particular dataset (e.g. how many spectra are available for averaging? Is it a DEC dataset? How high are fluxes?) the high frequency correction can either be calculated from the co-spectral data directly or has to be based on a-priori information (e.g. an estimation of damping timescales).

*Reply:* innFLUX calculates and stores cospectral information for the correction of cospectral attenuation. Which approach for such corrections is suitable or whether there exists a meaningful way for spectral corrections at all cannot be decided without further consideration of the user. The user must take into account measurement station geometry and operational parameters, distribution of sources and roughness elements as well as length and quality of the dataset for their decision whether and how to correct for cospectral loss of covariance. They have to validate the

respective assumptions of their approach (e.g. cospectral similarity of sensible heat flux and VOC flux; does the data set allow to come up with an adequate model spectrum? Are geometry and the operational parameters sufficiently well defined to determine all significant transfer functions?) for their specific data set, the measurement site as well as the setup and operation of the EC system and its sensors. We therefore believe the specific situation needs to undergo thorough evaluation by the experimentalist. We acknowledge that the reader needs guidance how to use the cospectral information provided by innFLUX. Thus we conducted a thorough analysis of the cospectral behavior of the VOC EC system described here based on Foken et al. (2012a) and now provide procedures and results in the manuscript.

*Changes:* See reply to comment 4 for details.

---

## Author Comment (AC2) · 20 Dec 2019

**Comment 1:** 'Unified' may not be an appropriate term for the title. This reviewer agrees in general with the assumptions and corrections included in the proposed methodology, but not all researchers may do it. Some debate exists on how to post process turbulent fluxes

*Reply and Changes:* We have changed unified to ' An open source code'. We also agree that there is some debate how many post processing steps are needed to arrive at accurate data. Especially small corrections that are prone to large uncertainties might not always be well constrained. We now include a thorough cospectral analysis and perform high frequency corrections as suggested by reviewer 1 (see reply and changes regarding comment #4 by reviewer 1).

**Comment 2:** A flowchart will help to visualize the order of the steps for post processing disjunct fluxes.

*Reply and Changes:* The revised manuscript now includes a flowchart that outlines individual work steps for processing EC data in Chapter S1 and Figure S1.

**Comment 3 :** The introduction should explain the need for measuring fluxes by eddy covariance over urban surfaces, particularly of speciated VOCs. Velasco et al. (2005, doi:10.1029/2005GL023356) deployed by first time a PTR-MS for measuring turbulent fluxes over an urban surface using the disjunct eddy covariance method. Some of the corrections and assumptions discussed here were also discussed by them.

*Reply and Changes:* We incorporated more discussion on the need to perform EC and also added new references as suggested.

**Comment 4:** "...and disjunct eddy covariance flux data."

*Reply:* OK Gibt's im manuscript nun in Sachen DEC eine reference zu Valescos Arbeit – ich finds nicht

**Comment 5:** Define acronyms every time they are used by first time. Please check this through-out the whole text. Many acronyms were used without being properly defined.

*Reply:* Ok we doublechecked and corrected acronyms throughout the text.

**Comment 6**: What about the met data necessary to compute turbulent fluxes?

*Reply:* Thank you for pointing out this omission. We included it in the abstract. The sentence now reads "We demonstrate the capabilities of the code based on a large urban dataset collected in Innsbruck, Austria, where three dimensional winds and ambient concentrations of NMVOC and auxiliary trace gases were sampled with high temporal resolution above an urban canopy."

**Comment 7:** Replace "surface fluxes" by "turbulent fluxes".

*Reply:* ok this was changed. Now the sentence reads: "Eddy covariance (EC) is the method of choice for most micrometeorological studies of turbulent fluxes (e.g. Dabberdt et al., 1993; Aubinet et al., 2012)."

**Comment 8:** Use italic fonts for referring to variables. Check this throughout the whole text.

*Reply:* Ok we corrected this.

**Comment 9:** in the horizontal dimension?

*Reply:* We rewrote this paragraph (in the new manuscript on P2L3+) completely according to suggestions from reviewer 1 (see also reply and changes regarding comment #3 by reviewer 1).

**Comment 10:** fast and highly accurate sensors

*Reply:* We changed this accordingly and the sentence now reads: "In the past EC has been largely restricted to a limited number of species (e.g. $H_2O$, $CO_2$, $CH_4$) due to the requirement of fast and highly accurate sensors (ideally sampling frequencies > 10Hz)."

**Comment 11:** Consider that the atmospheric reactivity of some species limits the application of the eddy covariance method for measuring turbulent fluxes. Some species react faster than the time taken by the air sample to reach the height of the instrumented tower. This is a particular constraint in polluted urban atmospheres.

*Reply:* We acknowledge the reviewer's comment on this issue. For typical heights measured in Innsbruck (approx. 40m above street level) we note that in our case it is only the interconversion between NO, $NO_2$ and $O_3$, that would warrant a significant consideration of reactivity. One of the fastest reacting VOC (e.g. Isoprene) has typical lifetimes of 30 minutes. The turbulent exchange time at our site is on the order of 200s. Thus a significant chemical

loss can be excluded for NMVOC reported in this manuscript. It is true though that this issue could become more important for measurements on tall towers where the vertical exchange time scale is much longer.

**Changes** We added a discussion on the issue of reactivity in the introduction.

**Comment 12:** Why is the co-spectral analysis important? What does it show?

**Reply:** We performed a thorough cospectral analysis based on Foken et al (2012a) and Lee et al. (2004) in response to this comment and comment #4 by reviewer 1. Such an analysis is important because it may allow for corrections of fluxes that are underestimated due to cospectral attenuation. In our example for toluene it shows that high frequency loss due to sensor separation, sonic path averaging, sensor averaging and tube attenuation is on average 2%.

**Changes:** We conducted a thorough analysis of the cospectral behavior of the VOC EC system described here based on Foken et al. (2012a) and Lee et al. (2004). The added Chapter S4 in the supplement now guides the reader how to derive a model cospectrum from quality checked individual half-hour cospectra (example in Figure S3). It shows how to determine transfer functions describing high frequency losses due to sensor separation, sonic path averaging, sensor path averaging (PTR-MS response) and tube attenuation, and how these attenuations cause loss of cospectral density at high frequencies, thus underestimating the flux (Figure 7). The new Chapter 3.5 in the main text points out the cospectral information calculated and stored by innFLUX, mentions both the experimental approach and the theoretical approach for the correction of high frequency losses, gives the user guidance which approach might be more appropriate, and details the procedure (reference to Chapter S4 and Figure S3) and results (Figure 7 and Figure S4) of the cospectral analysis.

**Comment 13:** The averaging time period depends also on the roughness elements' height. For flux measurements over smooth surfaces such as lakes and grasslands, for example, averaging time periods of 10-15 min are used, while for measurements over tall canopies in forested and urban environments, averaging periods of 30 min are common.

**Reply and Changes:** We agree and have added the clarification in the manuscript. Now the text reads: "For flux measurements over smooth surfaces such as lakes or short grasslands, averaging time periods as low as 15 min can be used. For measurements over tall canopies in forested and urban environments, it has been shown that 30 min averaging intervals are quite suitable for surface layer measurements, and that averaging periods up to 1 h can be feasible. Longer averaging periods often suffer from non-stationary conditions (Foken et al., 2010). "

**Comment 14:** Define inertial subrange

**Reply and Changes:** In context of turbulent kinetic energy (TKE), the inertial subrange is defined as the part of the co-spectrum where the energy density drops exponentially. The revised sentence reads: "A slow sensor will act as a low pass filter, where for example eddies in the inertial subrange (i.e. the co-spectral region where the energy density of the turbulent kinetic energy drops exponentially) cannot be fully resolved anymore."

**Comment 15:** Explain how you reached this figure.

**Reply:** With reference to the particular line, we assume the question is related to Figure 2. It shows the increasing scatter due to random (white) noise, when only one half hour period compared to a cumulated dataset is used for lag time determination. Assuming the lag time does not change between the cumulated half hour intervals and each of the individual half hour intervals, the analysis of lag time becomes more accurate due to a reduction in random noise.

**Comment 16:** You could save the reader of searching in a second article to learn about the eddy covariance flux system used here as a test case. Provide at least the local climate zone, land cover, measurement height, mean roughness elements height and zero-plane displacement height.

**Reply:** we have added the requested information to section 2.1 to make the manuscript more readable.

**Comment 17:** How many VOC species and of which groups (i.e. olefins, aromatics, etc.)?

**Reply and Changes:** We included a more detailed description of the calibration in Chapter S3 as well as Table S1 with the VOC species in the supplementary information.

| compound | protonated parent ion | m/z |
|---|---|---|
| Methanol | $(CH_4O)H^+$ | 33.03350 |
| Acetonitrile | $(C_2H_3N)H^+$ | 42.03382 |
| Acetaldehyde | $(C_2H_4O)H^+$ | 45.03350 |
| Acetone | $(C_3H_6O)H^+$ | 59.04914 |
| DMS | $(C_2H_6S)H^+$ | 63.02629 |
| Methyl-Ethyl-Ketone | $(C_4H_8O)H^+$ | 73.06480 |
| Benzene | $(C_6H_6)H^+$ | 79.05422 |
| 2-Methyl-3-buten-2-ol | $(C_5H_{10}O)H^+$ | 87.08045 |
| Toluene | $(C_7H_8)H^+$ | 93.06988 |
| m-Xylene | $(C_8H_{10})H^+$ | 107.08553 |
| 1,3,5-Trimethylbenzene | $(C_9H_{12})H^+$ | 121.10118 |

| 1,2,4,5-Tetramethylbenzene | (C10H14)H+ | 135.11683 |
| α-Pinene | (C10H16)H+ | 137.13248 |

**Comment 18:** What about data from a low frequency met sensor for flux corrections. The sonic/virtual temperature is not the absolute temperature. In most urban environments moisture in the air is inherent

*Reply:* Currently we apply corrections to the fast data stream. The sonic temperature is directly corrected by the instantaneous 10Hz $H_2O$ data (see Foken et al 2012a, eq. 4.1). In the case when no fast $H_2O$ data are available the code will not apply any corrections, and the user would need to apply an estimate of the Bowen ratio in the post-processing analysis.

**Comment 19:** But a co-spectra analysis is not feasible for DEC as explained above .....

*Reply:* In principle co-spectral analysis is always possible up to the Nyquist frequency. The ability to extract information for high frequency damping depends on whether the inertial subrange can be captured given a chosen DEC interval. This is location dependent (e.g. peak of the co-spectrum) and dependent on the DEC interval. In principle it should always be possible to perform co-spectral analysis in the low frequency domain.

**References**

Foken, Thomas, Ray Leuning, Steven R. Oncley, Matthias Mauder, and Marc Aubinet. "Corrections and Data Quality Control." In Eddy Covariance: A Practical Guide to Measurement and Data Analysis. Edited by Marc Aubinet, Timo Vesala, and Dario Papale. Dordrecht: Springer Netherlands, 2012a. DOI: 10.1007/978-94-007-2351-1_4

Foken et al., Coupling processes and exchange of energy and reactive and non-reactive trace gases at a forest site –results of the EGER experiment., Atm.Chem. Phys., 10.5194/acp-12-1923-2012, 2012

Karl, T., Guenther A., Lindinger C., Jordan A., Fall R., and Lindinger, W.: Eddy covariance measurements of oxygenated volatile organic compound fluxes from crop harvesting using a redesigned proton-transfer-reaction mass spectrometer, J. Geophys. Res., 106 (D20), 24157-24167, https://doi.org/10.1029/2000JD000112, 2001.

Karl, T. G., Spirig, C., Rinne, J., Stroud, C., Prevost, P., Greenberg, J., Fall, R., and Guenther, A.: Virtual disjunct eddy covariance measurements of organic compound fluxes from a subalpine forest using proton transfer reaction mass spectrometry, Atmos. Chem. Phys., 2, 279–291, https://doi.org/10.5194/acp-2-279-2002, 2002.

Lee, X., W.J. Massman, and B.E. Law. Handbook of Micrometeorology: A Guide for Surface Flux Measurement and Analysis: Kluwer Academic, 2004. http://books.google.com/books?id=IJ_19RkTfBQC.

Park et al., Active atmosphere ecosystem exchange of the vast majority of detected volatile organic compounds, Science, 10.1126/science.1235053, 2013.

Rantala et al., Anthropogenic and biogenic influence on VOC fluxes at an urban background site in Helsinki, Finland, Atm. Phys. Chem., 10.5194/acp-16-7981-2016, 2016

Rinne, H. J. I., Guenther, A. B., Warneke, C., de Gouw, J. A., and Luxembourg, S. L.: Disjunct eddy covariance technique for trace gas flux measurements, Geophys. Res. Lett., 28, 3139-3142, https://doi.org/10.1029/2001GL012900, 2001.

---

## Author Response (AR2)

Dear Eric,

We copyedited the manuscript regarding typos and formatting. The minor revisions are now incorporated.

Best regards,

Thomas